# Congenital Deletion of *Nedd4-2* in Lung Epithelial Cells Causes Progressive Alveolitis and Pulmonary Fibrosis in Neonatal Mice

**DOI:** 10.3390/ijms22116146

**Published:** 2021-06-07

**Authors:** Dominik H. W. Leitz, Julia Duerr, Surafel Mulugeta, Ayça Seyhan Agircan, Stefan Zimmermann, Hiroshi Kawabe, Alexander H. Dalpke, Michael F. Beers, Marcus A. Mall

**Affiliations:** 1Department of Pediatric Respiratory Medicine, Immunology and Critical Care Medicine, Charité-Universitätsmedizin Berlin, Corporate Member of Freie Universität Berlin and Humboldt-Universität zu Berlin, Augustenburger Platz 1, 13353 Berlin, Germany; dominik.leitz@charite.de (D.H.W.L.); marcus.mall@charite.de (M.A.M.); 2Translational Lung Research Center (TLRC), Member of the German Center for Lung Research (DZL), Department of Translational Pulmonology, University of Heidelberg, Im Neuenheimer Feld 156, 69120 Heidelberg, Germany; aycaseyhanagircan@med.uni-heidelberg.de; 3German Center for Lung Research (DZL), Associated Partner Site, Augustenburger Platz 1, 13353 Berlin, Germany; 4Division of Pulmonary, Allergy, and Critical Care Medicine, Perelman School of Medicine, University of Pennsylvania, 3450 Hamilton Walk Suite 216, Philadelphia, PA 19104, USA; mulugeta@pennmedicine.upenn.edu (S.M.); mfbeers@pennmedicine.upenn.edu (M.F.B.); 5Department of Infectious Diseases, Medical Microbiology and Hygiene, University of Heidelberg, 69120 Heidelberg, Germany; stefan.zimmermann@med.uni-heidelberg.de; 6Department of Molecular Neurobiology, Max Planck Institute of Experimental Medicine, Hermann-Rein-Str. 3D, 37075 Goettingen, Germany; kawabe@em.mpg.de; 7Laboratory of Molecular Life Science, Department of Gerontology, Institute of Biomedical Research and Innovation, Foundation for Biomedical Research and Innovation at Kobe, 2-2 Minatojima-Minamimachi Chuo-ku, Kobe 650-0047, Japan; 8Division of Pathogenetic Signaling, Department of Biochemistry and Molecular Biology, Kobe University Graduate School of Medicine, 1-5-6 Minatojima-minamimachi, Chuo-ku, Kobe 650-0047, Japan; 9Department of Pharmacology, Gunma University Graduate School of Medicine, 3-39-22 Showa-machi, Maebashi, Gunma 371-8511, Japan; 10Institute of Medical Microbiology and Virology, Medical Faculty, University Hospital Carl Gustav Carus Dresden, Technische Universität Dresden, 01307 Dresden, Germany; alexander.dalpke@ukdd.de; 11Berlin Institute of Health, Charité—Universitätsmedizin Berlin, Charitéplatz 1, 10117 Berlin, Germany

**Keywords:** Nedd4-2, animal model, interstitial lung disease, chILD, ENaC, proSP-C

## Abstract

Recent studies found that expression of NEDD4-2 is reduced in lung tissue from patients with idiopathic pulmonary fibrosis (IPF) and that the conditional deletion of *Nedd4-2* in lung epithelial cells causes IPF-like disease in adult mice via multiple defects, including dysregulation of the epithelial Na^+^ channel (ENaC), TGFβ signaling and the biosynthesis of surfactant protein-C proprotein (proSP-C). However, knowledge of the impact of congenital deletion of *Nedd4-2* on the lung phenotype remains limited. In this study, we therefore determined the effects of congenital deletion of *Nedd4-2* in the lung epithelial cells of neonatal doxycycline-induced triple transgenic *Nedd4-2^fl/fl^/CCSP-rtTA2^S^-M2/LC1* mice, with a focus on clinical phenotype, survival, lung morphology, inflammation markers in BAL, mucin expression, ENaC function and proSP-C trafficking. We found that the congenital deletion of *Nedd4-2* caused a rapidly progressive lung disease in neonatal mice that shares key features with interstitial lung diseases in children (chILD), including hypoxemia, growth failure, sterile pneumonitis, fibrotic lung remodeling and high mortality. The congenital deletion of *Nedd4-2* in lung epithelial cells caused increased expression of *Muc5b* and mucus plugging of distal airways, increased ENaC activity and proSP-C mistrafficking. This model of congenital deletion of *Nedd4-2* may support studies of the pathogenesis and preclinical development of therapies for chILD.

## 1. Introduction

Nedd4-2 is an E3 ubiquitin-protein ligase that participates in the posttranscriptional regulation of several proteins including ENaC, Smad2/3 and proSP-C, which play key roles in multiple cellular processes such as epithelial ion and fluid transport, TGFβ signaling and surfactant biogenesis that are essential for epithelial homeostasis and lung health [1,2,3,4,5,6,7,8]. In a previous study, we found that NEDD4-2 is reduced in the lung tissue of patients with idiopathic pulmonary fibrosis (IPF) [9]. Further, we demonstrated that the conditional deletion of *Nedd4-2* in lung epithelial cells by doxycycline induction of adult *Nedd4-2^fl/fl^/CCSP-rtTA2^S^-M2/LC1* mice, hereafter referred to as conditional *Nedd4-2^−/−^* mice, causes a chronic progressive, restrictive lung disease that shares key features with IPF in patients including signature lesions such as radiological and histological honeycombing and fibroblast foci [9]. These studies also identified the dysregulation of (i) ENaC, leading to airway surface liquid depletion and reduced mucociliary clearance; (ii) proSP-C biogenesis and (iii) TGFβ/Smad signaling, promoting fibrotic remodeling as epithelial defects and potential mechanisms triggering IPF-like disease in adult conditional *Nedd4-2^−/−^* mice [9].

Compared to the detailed characterization of the functional consequences and resulting pulmonary phenotype produced by the conditional deletion of *Nedd4-2* in the lung epithelial cells of adult mice [9], current knowledge on the impact of the congenital deletion of *Nedd4-2* on the lung phenotype in neonatal mice remains limited. A mouse line with constitutive systemic deletion of *Nedd4-2* demonstrated that the majority of mice lacking *Nedd4-2* died during or shortly after birth and that survivors developed substantial neutrophilic inflammation in the lungs at the age of 3 weeks [10]. Subsequent studies in mice with constitutive lung-specific deletion of *Nedd4-2* using a “leaky” *Nedd4-2^fl/fl^*/*Sftpc-rtTA/Cre* triple transgenic system under the control of the surfactant protein C (*Sftpc*) promoter showed massive neutrophilic inflammation, aspects of cystic fibrosis-like lung disease and premature death 3–4 weeks after birth [11]. However, the lung phenotype of neonatal *Nedd4-2^fl/fl^/CCSP-rtTA2^S^-M2/LC-1* mice, facilitating “tight” deletion of *Nedd4-2* in alveolar type 2 (AT2) cells as well as club cells of the conducting airways under control of the club cells 10 kDa secretory protein (*CCSP*) [12] promoter, has not been studied.

The aim of the present study was therefore to determine the effects of congenital deletion of *Nedd4-2* in lung epithelial cells of neonatal *Nedd4-2^fl/fl^/CCSP-rtTA2^S^-M2/LC-1* mice, hereafter referred to as congenital *Nedd4-2^−/−^* mice. Using physiologic, histopathologic, inflammatory and microbiological endpoints, we focused on the clinical phenotype including survival, lung morphology, inflammation markers in BAL, mucin (*Muc5b* and *Muc5ac*) expression in whole lung and airway mucus content, ENaC-mediated Na^+^ transport in freshly excised tissues of the conducting airways and proSP-C trafficking in AT2 cells to provide a comprehensive characterization of the lung phenotype of congenital *Nedd4-2^−/−^* mice, and to elucidate the impact of epithelial defects identified in adult conditional *Nedd4-2^−/−^* mice in the neonatal lung. The results of this study validate a new mouse model that shares key aspects of interstitial lung diseases in children (chILD), and thus offers new opportunities for studies of the pathogenesis and therapy of these childhood lung diseases with high unmet need [13].

## 2. Results

### 2.1. Congenital Deletion of Nedd4-2 in Lung Epithelial Cells Causes Severe Hypoxemia, Failure to Thrive and Early Mortality in Neonatal Mice

To determine the effect of the congenital deletion of *Nedd4-2* in epithelial cells of the neonatal mouse lung, we crossed mice carrying *Nedd4-2* flanked by *loxP* sites (*Nedd4-2^fl/fl^*) with *CCSP-rtTA2^S^-M2^S^/LC1* mice to enable tight doxycycline-dependent Cre expression for the targeted deletion of *Nedd4-2* in club cells of the conducting airways and AT2 cells of the lung [9,12]. Dams were continuously fed with doxycycline from the first day of mating to obtain triple transgenic congenital *Nedd4-2^−/−^* mice. At 10 days after birth, before the onset of clinical signs of lung disease, body weight did not differ between congenital *Nedd4 2**^−/−^* mice (5.4 ± 0.08 g) vs. littermate controls (5.3 ± 0.11 g). Around 3 weeks after birth, congenital *Nedd4-2^−/−^* mice showed clinical symptoms of respiratory distress with severe hypoxemia (Figure 1a), weight loss (Figure 1b) and ~95% mortality within 4 weeks after birth (Figure 1c).

### 2.2. Congenital Deletion of Nedd4-2 in Lung Epithelial Cells Causes Alveolar Inflammation and Fibrosis in Neonatal Mice

Microscopically, hematoxylin- and eosin (H&E) stained lung sections from 10-day-old congenital *Nedd4-2^−/−^* mice did not show abnormalities compared to littermate controls (Figure 2a), whereas lung sections from 3-week-old congenital *Nedd4-2^−/−^* mice displayed patchy inflammatory infiltrates, especially in the periphery of the lung (Figure 2b). These same regions also showed evidence of epithelial hyperplasia and alveolitis, with large foamy macrophages and granulocytes infiltrating the alveolar airspaces in the affected areas (Figure 2b). Masson-Goldner-Trichrome staining of lung sections of 3-week-old congenital *Nedd4-2^−/−^* mice showed substantial collagen deposition in affected lung regions (Figure 2c). The use of multiple control lines established that the observed phenotype was not caused by off-target effects of rtTA, Cre recombinase or doxycycline and that the expression system was tight in the absence of doxycycline (Figure A1, Appendix A).

### 2.3. Development of Pneumonitis in Congenital Nedd4-2^−/−^ Mice

BAL studies demonstrated that the histological pneumonitis observed in congenital *Nedd4-2^−/−^* mice was accompanied by a dynamic polycellular inflammatory cell influx, as well as a mixed proinflammatory cytokine response. Assaying the BAL of 10-day-old and 3-week-old congenital *Nedd4-2^−/−^* mice revealed that the congenital deletion of *Nedd4-2^−/−^* produced an early (10 days) increase in the number of macrophages that demonstrated morphologic features of activation, including irregular shape, vacuolized cytoplasm and increased size, which was accompanied by increased acitiviy of matrix-metalloproteinase 12 (Mmp12) on the cell surface (Figure 3a–d), as previously described in *Scnn1b*-Tg mice with muco-obstructive lung disease [14,15]. Inflammatory parameters further increased by the age of 3 weeks, with elevated numbers of neutrophils and eosinophils (Figure 3e,f), as well as increased concentrations of KC, IL-13 and IL-1β (Figure 3g–i).

The observed inflammation in congenital *Nedd4-2^−/−^* mice was not attributable to bacterial infection. Microbiological surveys of BAL of 3-week-old congenital *Nedd4-2^−/−^* mice and littermate controls using bacterial cultures (Figure 4a,b), as well as 16S rRNA PCR (Figure 4c), did not show any evidence of bacterial infection.

### 2.4. Congenital Deletion of Nedd4-2 in Lung Epithelial Cells Causes Mucus Plugging and Epithelial Necrosis in Distal and Terminal Airways in Neonatal Mice

Previous studies in adult mice with conditional deletion of *Nedd4-2* identified epithelial remodeling of the distal airways with increased numbers of mucin-producing goblet cells, expression of Muc5b and impaired mucociliary clearance as key features of IPF-like lung disease in this model [9]. We therefore determined expression of the secreted mucins Muc5b and Muc5ac and mucus content in lungs of congenital *Nedd4-2^−/−^* mice. Transcript levels of Muc5b and Muc5ac were increased in the lungs of 3-week-old congenital *Nedd4-2^−/−^* mice compared to controls (Figure 5a,b). Alcian blue-periodic acid–Schiff (AB-PAS) staining of lung sections showed goblet cell metaplasia and mucus plugging in the distal and terminal airways of 3-week-old congenital *Nedd4-2^−/−^* mice, especially in regions with a high grade of inflammation and fibrosis (Figure 5c), but not in age-matched littermate controls. Previous studies in *Scnn1b*-Tg mice and patients with muco-obstructive lung disease demonstrated that airway mucus plugging, probably via local hypoxia in the airway lumen, led to hypoxic degeneration and necrosis of airway epithelial cells [16,17,18,19]. Similarly, we found increased numbers of degenerative cells in the mucus-obstructed distal and terminal airways, especially in inflamed and fibrotic lung regions of 3-week-old congenital *Nedd4-2^−/−^* mice (Figure 5d,e).

### 2.5. Increased ENaC Activity in Freshly Excised Airway Tissues of Congenital Nedd4-2^−/−^ Mice

Nedd4-2 was shown to regulate cell surface expression of ENaC [4,20], and our previous studies demonstrated that a lack of Nedd4-2 caused increased ENaC function, which led to airway surface liquid depletion and impaired mucociliary clearance in adult conditional *Nedd4-2^−/−^* mice [9]. To investigate the effects of the congenital deletion of Nedd4-2 on ENaC activity, we performed bioelectric Ussing chamber experiments in freshly excised tracheal tissue from 10-day-old neonatal mice. At postnatal day 10, i.e., prior to detectable histological changes, the ENaC-mediated amiloride-sensitive short circuit current (I_SC_) was significantly increased in congenital *Nedd4-2^−/−^* mice compared to littermate controls (Figure 6a,b), supporting a role of increased ENaC acitvity in the patohphysiology of the observed phenotype.

### 2.6. proSP-C Is Mistrafficked in Lung Epithelial Cells of Congenital Nedd4-2^−/−^ Mice

Nedd4-2 was also shown to play a role in the posttranslational regulation of SP-C expressed in AT2 cells, and previous studies found mutations in the SFTPC gene in association with the development of ILD both in children (chILD) and in familial IPF in adults [21,22,23,24,25,26]. In our previous studies, we found that a lack of Nedd4-2 causes mistrafficking of proSP-C, but that this defect did not play a dominant role in determining the IPF-like lung phenotype produced by the conditional deletion of *Nedd4-2* in adult mice [9]. To determine the impact of proSP-C mistrafficking due to lack of Nedd4-2 in the neonatal lung, we performed biochemical studies and investigated the effect of the genetic deletion of Sftpc in congenital *Nedd4-2^−/−^* mice.

Using double label fluorescence immunohistochemistry for proSP-C and Lamp-1, we found that, in 3-week-old neonatal control mice, the subcellular distribution of proSP-C was predominantly found in Lamp-1 positive lamellar bodies. In the lungs of 3-week-old congenital *Nedd4-2^−/−^* mice, similar to our findings in adult conditional *Nedd4-2^−/−^* mice [9], a significant proportion of proSP-C expression shifts to Lamp-1 negative cytosolic compartments (Figure 7a). The mistrafficking of proSP-C was accompanied by marked changes in its posttranslational processing. Western blots of proSP-C from lung homogenates of 3-week-old mice revealed a 21–22 kDa proSP-C doublet in control mice while, in congenital *Nedd4-2^−/−^* mice, the primary translation product doublet shifts to a single band, accompanied by the appearance of a new intermediate around 16 or 17 kDa (Figure 7b). In BAL, Western blotting revealed a reduction in mature SP-C in 3-week-old congenital *Nedd4-2^−/−^* mice compared to littermate controls (Figure 7c). Despite a major impact on SP-C biosynthesis, other components of the surfactant system, such as surfactant protein B and D (SP-B and SP-D), were largely unaffected (Figure 7b).

Despite in vivo confirmation of the previously described role for NEDD4-2 in SFTPC biosynthesis [1,2], and similar to our previous studies in adult conditional *Nedd4-2^−/−^* mice [9], we found that proSP-C mistrafficking alone was insufficient to drive the abnormal lung phenotype found in neonatal mice with the congenital deletion of *Nedd4-2*. When *Nedd4-2^fl/fl^/CCSP-rtTA2^S^-M2/LC1* mice were crossed with *Sftpc*-deficient (*Sftpc^−/−^*) mice and induced in utero with doxycycline, the genetic deletion of *Sftpc* in quadruple transgenic mice had no effect on survival (Figure 7d), the number of BAL macrophages (Figure 7e), neutrophils (Figure 7f), eosinophils (Figure 7g) or on structural lung disease (data not shown) compared to triple transgenic congenital *Nedd4-2^−/−^* mice. These data are consistent with our previous results in adult conditional *Nedd4-2^−/−^* mice, and imply that congenital *Nedd4-2* deficiency imparts a toxic effect that is not attributable to a single protein but more likely caused by pleiotropic effects on AT2 cell homeostasis.

## 3. Discussion

This study demonstrates that the congenital deletion of *Nedd4-2* in lung epithelial cells causes a spontaneous and rapidly progressive lung disease in neonatal mice that shares key clinical and histopathological features of interstitial lung diseases in children (chILD), and thereby extends recent reports on the E3 ubiquitin ligase NEDD4-2 in the pathogenesis of ILD [9]. These features include respiratory distress, hypoxemia, growth failure, sterile alveolitis, patchy fibrotic remodeling of the alveolar airspaces and high neonatal mortality (Figure 1, Figure 2, Figure 3 and Figure 4) [27,28]. Similar to conditional deletion in adult mice [9], we found that the congenital deletion of *Nedd4-2* results in increased expression of the mucins *Muc5b* and *Muc5ac* and a remodeling of the distal airways including goblet cell metaplasia in congenital *Nedd4-2^−/−^* mice (Figure 5). In addition, epithelial defects previously reported in adult conditional *Nedd4-2^−/−^* mice, such as increased ENaC-mediated Na^+^/fluid transport and abnormal proSPC trafficking, were confirmed in the lungs of neonatal congenital *Nedd4-2^−/−^* mice (Figure 6 and Figure 7) [9]. Taken together, these results demonstrate that Nedd4-2 in lung epithelial cells plays an important role in normal lung development, provide additional evidence for its importance in lung health and have established a mouse model of chILD, comprising a spectrum of lung diseases in children with high unmet need.

Besides the important similarities of pulmonary phenotypes caused by the congenital vs. the conditional deletion of *Nedd4-2* in the murine lung, including restrictive lung disease with patchy fibrotic remodeling of distal airspaces due to dysregulated Smad2/3 signaling, leading to increased levels of TGFβ, remodeling of distal airways with goblet cell metaplasia and increased expression of *Muc5b*, as well as high pulmonary mortality (Figure 1, Figure 2 and Figure 5) [9,29], our study also revealed some striking age-dependent differences. First, the onset and progression of ILD was substantially accelerated in congenital vs. conditional *Nedd4-2^−/−^* mice, as evidenced by the time point of mortality that occurred within ~4 weeks after birth in most neonatal congenital *Nedd4-2^−/−^* mice compared to ~4 months after conditional deletion of *Nedd4-2^−/−^* in adult mice (Figure 1) [9]. Second, alveolitis with inflammatory cell infiltrates, including morphologically activated “foamy” macrophages, neutrophils and eosinophils associated with elevated pro-inflammatory cytokines such as IL-1β, KC and IL-13 in BAL, was substantially more prominent in neonatal congenital *Nedd4-2^−/−^* compared to conditional *Nedd4-2^−/−^* mice (Figure 3) [9]. Third, histopathologic studies of the lungs of congenital *Nedd4-2^−/−^* mice revealed mucus plugging of the distal airways that was associated with hypoxic epithelial necrosis (Figure 5), a phenotype that was previously reported in neonatal *Scnn1b*-Tg mice with muco-obstructive lung disease [16,17,18,19], but not observed in adult conditional *Nedd4-2^−/−^* mice [9].

Based on these findings, we studied the role of pro-SPC trafficking in AT2 cells and ENaC-mediated Na^+^ transport across freshly excised airway tissues of neonatal congenital *Nedd4-2^−/−^* mice, i.e., epithelial cell functions that we previously found to be abnormal in adult conditional *Nedd4-2^−/−^* mice, as a potential explanation for these age-dependent differences in lung phenotypes. Using a variety of techniques, our data provide evidence of defective proSP-C trafficking, maturation and secretion in this neonatal model (Figure 7), which parallels findings we reported in adult conditional *Nedd4-2^−/−^* mice [9]. However, similar to adult conditional *Nedd4-2^−/−^* mice, the genetic deletion of *Sftpc* was insufficient to rescue the lung disease phenotype in congenital *Nedd4-2^−/−^* mice (Figure 7) [9]. Thus, the effect size of neither misprocessed proSP-C nor loss of mature SP-C in surfactant is sufficient to drive the ILD phenotype and explain the age-dependent differences observed in neonatal congenital vs. adult conditional *Nedd4-2^−/−^* mice.

Similar to previous studies in adult conditional *Nedd4-2^−/−^* mice [9], we show that congenital deletion of *Nedd4-2* produces increased ENaC activity in airway epithelial cells of neonatal mice (Figure 6). In adult conditional *Nedd4-2^−/−^* mice, we demonstrated that increased ENaC-mediated Na^+^/fluid absorption across airway epithelia, as previously shown in patients with cystic fibrosis and *Scnn1b*-Tg mice [6,30,31,32,33], results in airway surface liquid depletion and impaired mucociliary clearance [9]. As mucociliary clearance is an important innate defense mechanism of the lung, and retention of inhaled irritants and pathogens leads to repeated micro-injury and chronic inflammation, our data support mucociliary dysfunction as an important disease mechanism triggering ILD in both congenital and adult conditional *Nedd4-2^−/−^* mice [9,32,34,35]. Of note, this concept is consistent with studies in *Muc5b*-overexpressing mice that exhibit impaired mucociliary clearance and develop more severe bleomycin-induced pulmonary fibrosis [36]. The importance of dysregulated ENaC activity in the pathogenesis of ILD in congenital *Nedd4-2^−/−^* mice is also supported by the observation that this epithelial ion transport defect was already present in 10-day-old mice with normal lung morphology, i.e., prior to the onset of histological signs of ILD (Figure 2), as well as previous studies in *Nedd4-2^fl/fl^*/*Sftpc-rtTA/Cre* mice with the constitutive deletion of *Nedd4-2* under control of the SP-C promoter [11] and mice with the constitutive overexpression of the α and β subunits of ENaC in the lung [37]. In both models, increased ENaC activity in the distal lung was associated with severe pulmonary inflammation, mucus obstruction of distal airways and high neonatal mortality [11,37]. Taken together, these data support increased ENaC activity leading to airway/alveolar surface liquid depletion and mucociliary dysfunction in distal airways as a key pathogenetic mechanism of ILD in congenital *Nedd4-2^−/−^* mice.

Interestingly, a previous study in fetal distal lung epithelial cells of wild-type rats found that the male sex is associated with reduced ENaC-mediated Na^+^ transport [38]. Our study included all newborns from each litter, resulting in a balanced distribution of male and female neonates that enabled an exploratory analysis of potential gender differences. Similar to previous studies in rat lung epithelia [38], we observed a ~30% reduction in ENaC-mediated Na^+^ absorption in male vs. female mice in the control group, as well as the congenital *Nedd4-2^−/−^* group (data not shown). However, this gender difference in ENaC function did not reach statistical significance based on the number of mice available for our study. Similar, other pulmonary phenotypes of neonatal congenital *Nedd4-2^−/−^* mice including hypoxemia, growth failure, pulmonary inflammation, mucin expression, epithelial cell necrosis, abnormal proSP-C trafficking and mortality did not differ between male vs. female mice. However, our study was not powered to detect gender differences, and future studies are necessary to determine the potential role of gender differences in ENaC-mediated Na^+^ absorption in the pathogenesis of lung disease in congenital *Nedd4-2^−/−^* mice.

Several factors may explain the age-specific differences in pulmonary phenotypes produced by deletion of *Nedd4-2* in neonatal vs. adult mice. First, the accelerated onset and increased severity of pulmonary inflammation observed in congenital *Nedd4-2^−/−^* mice may be explained by an increased susceptibility of the neonatal lung to the retention of inhaled irritants, as previously shown for cigarette smoke exposure in *Scnn1b*-Tg mice with muco-obstructive lung disease [39]. Second, in congenital *Nedd4-2^−/−^* mice, we found that increased ENaC activity leading to mucociliary dysfunction, probably due to a smaller diameter of neonatal vs. adult airways, is associated with mucus plugging and hypoxic epithelial cell necrosis of the distal airways (Figure 5), whereas this phenotype was not observed in conditional *Nedd4-2^−/−^* mice [9]. As hypoxic epithelial cell necrosis in mucus-obstructed airways has been identified as a strong trigger of sterile inflammation via triggering the pro-inflammatory IL-1 signaling pathway in the absence of bacterial infection in *Scnn1b*-Tg mice, and patients with muco-obstructive lung diseases such as cystic fibrosis and chronic obstructive pulmonary disease [16,40,41,42], this mechanism may also contribute to the more severe inflammatory phenotype caused by the congenital deletion of *Nedd4-2* in the neonatal lung. Finally, the differences in the onset and progression of ILD in congenital vs. conditional *Nedd4-2^−/−^* mice may be explained by age-dependent differences in the temporal and spatial activity of the CCSP promoter observed in previous studies of the *CCSP-rtTA2^S^-M2* activator line that was used for inducible lung-specific deletion of *Nedd4-2* [9,12]. These studies demonstrated a broader expression of the reverse tetracycline transactivator rtTA2^S^-M2 in AT2 cells, as well as club cells throughout the conducting airways of the neonatal lung whereas, in adult mice rtTA2^S^-M2 expression was more restricted to AT2 cells and club cells of the distal airways [12]. In addition, a previous study demonstrated age-dependent activity of the CCSP promoter, with the highest levels around birth and decreasing activity in older mice [43]. These temporal and spatial differences are expected to result in a faster and more widespread deletion of *Nedd4-2* in the neonatal vs. adult lung that may aggravate increased ENaC activity and mucociliary dysfunction, increased pro-fibrotic TGFβ signaling and potentially other pathogenic processes induced by *Nedd4-2* deficiency [1,2,3,8,9,10,11,44,45,46,47,48,49]; therefore, they might also contribute to the more rapid onset and progression of ILD in congenital vs. conditional *Nedd4-2^−/−^* mice.

Previous studies demonstrated that systemic deletion of *Nedd4-2* leads to perinatal lethality in mice and loss-of-function variants of *NEDD4-2* have not been described in humans [10]. In our study, targeted in utero deletion of *Nedd4-2* in lung epithelial cells did not cause perinatal morbidity or mortality, as evidenced by a normal distribution of genotypes and as expected from Mendelian ratios, normal development and weight gain, as well as a lack of respiratory symptoms in the the first 10 days of life (Figure 2, Figure 3 and Figure A2). However, our data demonstrate that the congenital deletion of *Nedd4-2* in the lung leads to an early onset and rapid progression of ILD beyond the perinatal period (Figure 1, Figure 2 and Figure 3). In our previous study, we found that NEDD4-2 protein and transcript levels were reduced in lung tissue biopsies from IPF patients, supporting the role of NEDD4-2 dysfunction in human ILD [9]. Based on these findings in adult IPF patients, we speculate that NEDD4-2 deficiency may also be implicated in the pathogenesis of chILD. However, future studies are necessary to test this hypothesis and determine mechanisms of lung-specific NEDD4-2 deficiency that may be caused, e.g., by transcriptional, post-transcriptional or epigenetic regulation of NEDD4-2 in the lung.

In summary, our results demonstrate that the congenital deletion of *Nedd4-2* in lung epithelial cells causes severe ILD in neonatal mice that shares key features with interstitial lung diseases in children (chILD), including respiratory distress, hypoxemia, growth failure, sterile alveolitis, progressive fibrotic remodeling of the lung parenchyma and high mortality. These data further substantiate an important role of Nedd4-2 in normal lung development and lung health, and have established a mouse model of chILD that may serve as a useful tool for studies of the complex in vivo pathogenesis, the identification of biomarkers and therapeutic targets, as well as preclinical evaluations of novel therapeutic strategies that are urgently needed to improve the clinical outcome of patients with chILD [13].

## 4. Materials and Methods

### 4.1. Experimental Animals

All animal studies were approved by the animal welfare authority responsible for the University of Heidelberg (Regierungspräsidium Karlsruhe, Karlsruhe, Germany). Mice for congenital deletion of *Nedd4-2* in lung epithelial cells were generated as previously described [9]. In brief, mice carrying *Nedd4-2^fl/fl^* [11] were intercrossed with *CCSP-rtTA2^S^-M2* line 38 (*CCSP-rtTA2^S^-M2*) [12] and *LC1* mice [50,51]. All three lines were on a C57BL6/N background. *Sftpc^−/−^* mice [52] were obtained on a 129S6 background. Mice were housed in a specific pathogen-free animal facility and had free access to food and water. For prenatal induction, dams were treated continuously with doxycycline from the first day of mating and mice were studied at 10 days and 3 weeks of age. All newborn mice of a litter were included in our study, irrespective of gender and genotype, yielding a balanced gender distribution in the control groups and congenital *Nedd4-2**^−/−^* groups. Details on the genotype distribution are provided in Figure A2, Appendix A.

### 4.2. Measurement of Inflammatory Markers in BAL

BAL was performed and differential cell counts and macrophage sizes were determined as previously described [17]. Concentrations of KC (CXCL-1) and IL-13 were measured in cell-free BAL supernatant and IL-1β was measured in total lung homogenates by ELISA (R&D Systems, Minneapolis, MN, USA) according to manufacturer’s instructions. Mmp12 acitivity on the surface of BAL macrophages was assessed by a Foerster resonance energy transfer (FRET) based activity assay as previously described [14]. In brief, BAL cells were incubated for 10 min at room temperature with the membrane-anchored FRET reporter Laree1 (1 µM). Cells were diluted with PBS to a volume of 200 µL and centrifuged on slides by cytospin. Membrane-bound Mmp12 activity was measured by confocal microscopy. Images were acquired on a Leica SP8 confocal microscope with an HC PL APO CS2 63× 1.3 oil objective (Leica microsystems, Wetzlar, Germany). Donor/acceptor ratio was calculated using the open source imaging analysis software Fiji version 1.46r [53,54].

### 4.3. Histology and Morphometry

Right lungs were inflated with 4% buffered formalin to 25 cm of fixative pressure. Non-inflated left lungs were immersion fixed. Lungs were paraffin embedded and sectioned at 5 µm and stained with H&E, Masson-Goldner-Trichrome and AB-PAS. Images were captured with a NanoZoomer S60 Slidescanner (Hamamatsu, Hamamatsu City, Japan) at a magnification of 40×. Airway regions were determined from proximal-to-distal distances and airway branching, as determined by longitudinal sections of lung lobes at the level of the main axial airway, as previously described [55]. Degenerative cells were identified by morphologic criteria such as swollen cells with vacuolized cytoplasm and pycnotic nucleus in H&E stained lung sections. Numeric cell densities were determined using NDP.view2 software version 2.7.52 (Hamamatsu, Hamamatsu City, Japan), as previously described [16].

### 4.4. Pulse Oximetry

Oxygen saturation of 3-week-old mice was determined using a noninvasive pulse oximeter for laboratory animals (MouseOx**^®^** Plus, Starr Life Science, Oakmont, PA, USA) and measured with a thigh clip sensor, as previously described [9]. Percent oxygen saturation was measured after stabilization of heart rate and breathing frequency.

### 4.5. Immunofluorescence Microscopy

Lung sections were evaluated for proSP-C using a primary polyclonal anti-NproSP-C antibody and Alexa Fluor 488 conjugated goat anti-rabbit IgG (Jackson Immuno Research, 111-545-062, West Grove, PA, USA), as described previously [24]. Confocal images were acquired using a 488 nm laser line package of an Olympus Fluoview confocal system attached to an Olympus IX81 microscope (60× oil objective).

### 4.6. SDS-PAGE and Immunoblotting

Sodium dodecyl sulfate polyacrylamide gel electrophoresis (SDS-PAGE) using Novex Bis–Tris gels (NP0301, ThermoFisher Scientific, Waltham, MA, USA) and immunoblotting of PVDF membranes with primary antisera followed by species specific horseradish peroxidase conjugated secondary antisera was performed as published [24,56]. Bands detected by enhanced chemiluminescence (ECL2, ThermoFisher Scientific, 80196, Waltham, MA, USA; or WesternSure, LI-COR, 926-95000, Lincoln, NE, USA) were acquired by exposure to film or direct scanning using an LI-COR Odyssey Fc Imaging Station (Lincoln, NE, USA) and quantitated using the manufacturer’s software. For immunoblotting of surfactant proteins, the following antisera were used. Polyclonal anti- NproSP-C raised against the Met [10]–Glu [23] domain of rat proSP-C peptide, polyclonal anti-SP-B (PT3) raised against purified bovine SP-B and polyclonal anti-SP-D (antisera 1754) raised against 2 synthetic SP-D peptides were each produced in rabbits in house and validated as published [56,57,58,59]. Polyclonal mature anti-SP-C antisera was obtained from Seven Hills Bioreagents (WRAB-76694; Cincinnati, OH, USA) and validated in a prior study [24]. Monoclonal anti-Actb was obtained from Sigma Aldrich (A1978) St. Louis, MO, USA.

### 4.7. Electrogenic Ion Transport Measurements

Mice were deeply anesthetized via intraperitoneal injection of a combination of ketamine and xylazine (120 mg/kg and 16 mg/kg, respectively) and killed by exsanguination. Airway tissues were dissected using a stereomicroscope as previously described [60,61] and immediately mounted into perfused micro-Ussing chambers. Experiments were performed at 37 °C under open-circuit conditions and amiloride-sensitive ENaC-mediated short circuit current (I_SC_) was determined as previously described [61].

### 4.8. mRNA Expression Analysis

Lungs from mice were stored at 4 °C in RNAlater (Applied Biosystems, Darmstadt, Germany). Total RNA was extracted using Trizol reagent (Invitrogen, Karlsruhe, Germany) according to manufacturer’s instructions. cDNA was obtained by reverse transcription of 1 μg of total RNA with Superscript III RT (Invitrogen, Karlsruhe, Germany). To analyze mRNA expression of mucins, quantitative real-time PCR was performed on an Applied Biosystems 7500 Real Time PCR System using TaqMan universal PCR master mix and the following inventoried TaqMan gene expression assays for *Muc5b* (Accession No. NM_028801.2; Taqman ID Mm00466391_m1) and *Muc5ac* (Accession No. NM_010844.1; Taqman ID Mm01276718_m1) (Applied Biosystems, Darmstadt, Germany) according to manufacturer’s instructions. Relative fold changes of target gene expression were determined by normalization to expression of the reference gene *Actb* (Accession No. NM_007393.1; Taqman ID Mm00607939_s1) [17,62].

### 4.9. Microbiology Studies

BAL was performed in 3-week-old mice under sterile conditions. Mice were deeply anesthetized via intraperitoneal injection with a combination of ketamine and xylazine (120 mg/kg and 16 mg/kg, respectively) and killed by exsanguination. A cannula was inserted into the trachea and whole lungs were lavaged 3 times with 300 μL PBS. The recovered BAL fluid was plated on columbia blood agar (Becton Dickinson, Heidelberg, Germany), chocolate agar, Mac Conkey agar, prereduced Schaedler agar and kanamycin-vancomycin blood agar plates (bioMérieux, Nürtingen, Germany). After 48 h of incubation at 37 °C, colony forming units were counted and classified by MALDI-TOF mass-spectrometry (Bruker Daltonik, Bremen, Germany). Then, 16S rRNA PCR was performed to detect non-culturable bacterial species [63].

### 4.10. Statistical Analysis

All data are shown as mean ± S.E.M. Data were analyzed with GraphPad Prism version 7 (GraphPad Software Inc, LaJolla, CA, USA). Distribution of data was assessed with Shapiro–Wilk test for normal distribution. For comparison of two groups, unpaired two-tailed t-test or Mann–Whitney test were used as appropriate. Comparison of more than two groups with normally distributed data was performed with one-way ANOVA followed by Tukey’s post hoc test. Genotype frequency was analyzed by χ^2^ test. Comparison of survival was evaluated using the log rank test. A *p* value < 0.05 was accepted to indicate statistical significance.

## Figures and Tables

**Figure 1 ijms-22-06146-f001:**
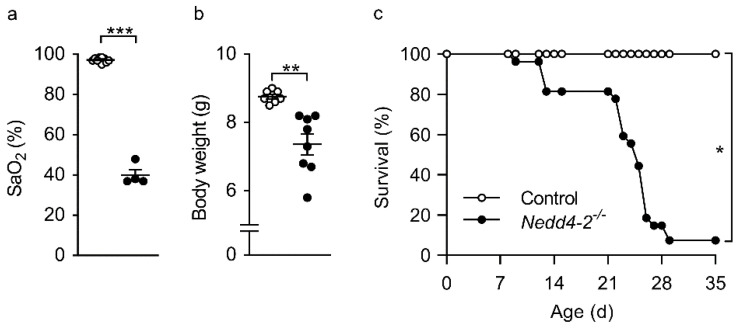
Congenital deletion of *Nedd4-2* in lung epithelial cells causes severe hypoxemia, failure to thrive and high mortality in neonatal mice: (**a**,**b**) Oxygen saturation (**a**) and body weight (**b**) measured in 3-week-old mice. *n* = 4–13 mice per group. ** *p* < 0.01, *** *p* < 0.001. (**c**) Survival curve of congenital *Nedd4-2^−/−^* and control mice. *n* = 18–27 mice per group. * *p* < 0.05. Data are shown as mean ± S.E.M.

**Figure 2 ijms-22-06146-f002:**
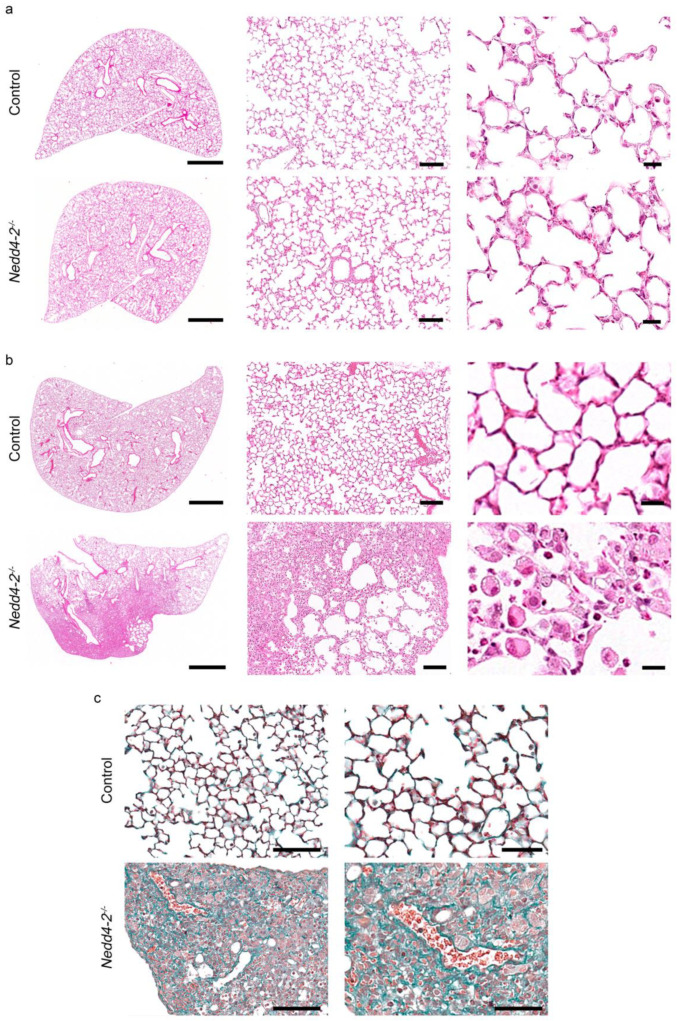
Congenital deletion of *Nedd4-2* in lung epithelial cells causes alveolar inflammation and fibrosis in neonatal mice: (**a**,**b**) Representative micrographs of H&E stained lung sections of 10-day-old (**a**) and 3-week-old (**b**) congenital *Nedd4-2^−/−^* and control mice. Scale bars, 1 mm (left column), 100 µm (middle column) and 15 µm (right column). (**c**) Masson-Goldner-Trichrome stained lung sections of 3-week-old congenital *Nedd4-2^−/−^* and control mice. Scale bars, 100 µm (left column) and 50 µm (right column). *n* = 7–10 mice per group.

**Figure 3 ijms-22-06146-f003:**
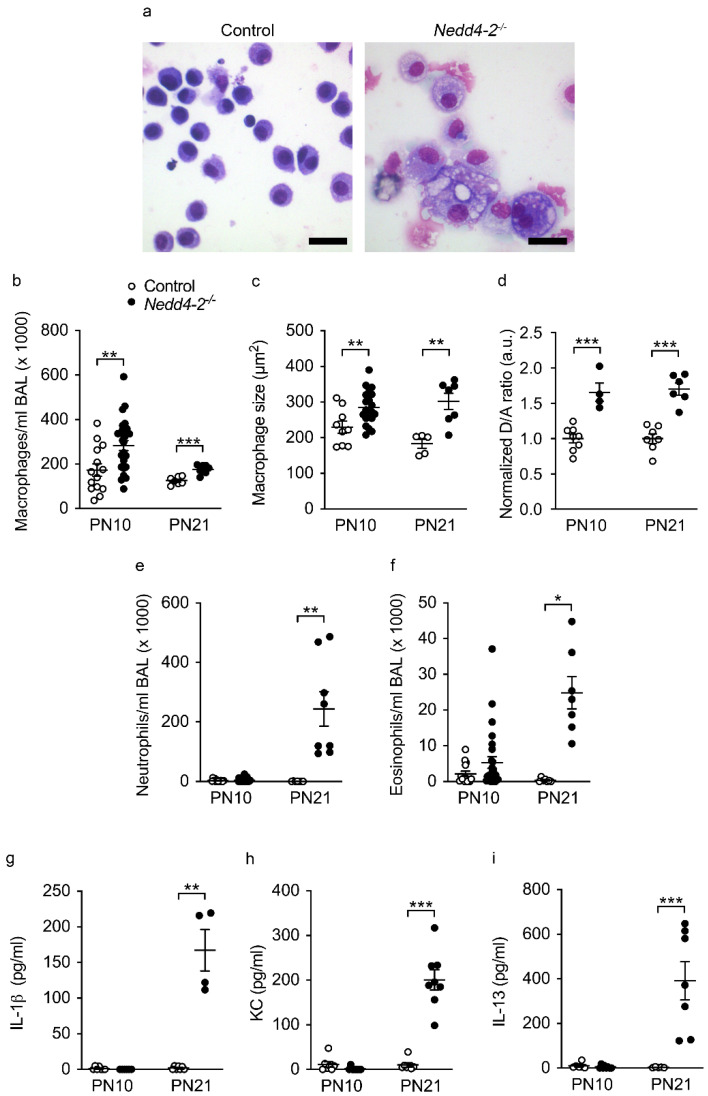
Development of pneumonitis in neonatal mice with congenital deletion of *Nedd4-2*: (**a**) Representative micrographs of BAL macrophages in 3-week-old congenital *Nedd4-2^−/−^* and control mice. Scale bars, 20 µm. (**b**,**c**) Number (**b**) and size (**c**) of macrophages in BAL. (**d**) Quantification of Mmp12 activity on the surface of BAL macrophages using the membrane-bound FRET reporter Laree1. Mmp12 activity was determined from the donor to acceptor (D/A) ratio of fluorescence emission produced by Mmp12-mediated cleavage of Laree1. Data were normalized to age-matched control mice. (**e**,**f**) Number of neutrophils (**e**) and eosinophils (**f**) in BAL of 10-day and 3-week-old congenital *Nedd4-2^−/−^* and control mice. (**g**–**i**) Concentrations of IL-1β (**g**), KC (**h**) and IL-13 (**i**) in BAL supernatant of 10-day and 3-week-old congenital *Nedd4-2^−/−^* and control mice. *n* = 4–27 animals per group. * *p* < 0.05, ** *p* < 0.01, *** *p* < 0.001. Data are shown as mean ± S.E.M.

**Figure 4 ijms-22-06146-f004:**
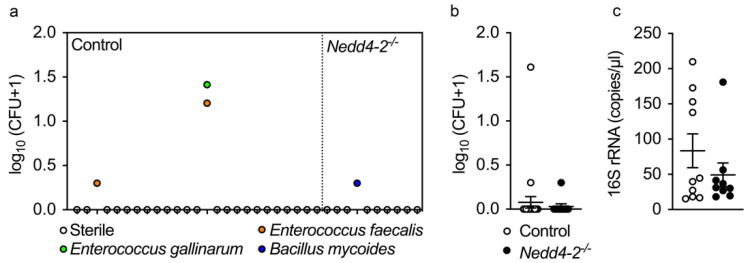
Bacterial species detected in BAL by bacterial culture and 16S rRNA quantitative PCR in neonatal mice with congenital deletion of *Nedd4-2* and littermate controls: (**a**) Individual colony forming units (CFUs) for bacterial species cultured from BAL of 3-week-old congenital *Nedd4-2^−/−^* mice and littermate controls. Each dot represents the result obtained from an individual mouse. (**b**) Summary of all CFUs in BAL of 3-week-old *Nedd4-2^−/−^* mice and littermate controls. (**c**) Bacterial load determined by 16S rRNA analysis in BAL. *n* = 10–25 mice per group. Data are shown as mean ± S.E.M.

**Figure 5 ijms-22-06146-f005:**
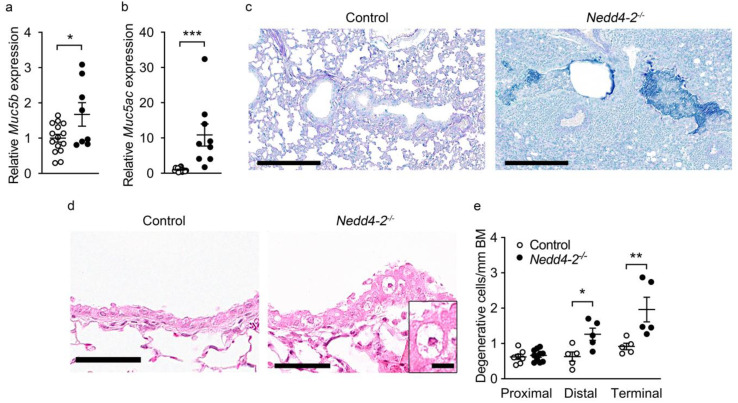
Congenital deletion of *Nedd4-2* in lung epithelial cells causes mucus plugging and epithelial necrosis in distal and terminal airways of neonatal mice: (**a**,**b**) mRNA expression levels of *Muc5b* (**a**) and *Muc5ac* (**b**) in whole lungs from congenital *Nedd4-2**^−/−^* and control mice. (**c**) Representative airway sections from 3-week-old congenital *Nedd4-2^−/−^* and control mice stained with AB-PAS to illustrate the presence of mucus. Scale bars, 200 µm. (**d**) Representative H&E stained lung sections of distal airways showing degenerative airway epithelial cells in 3-week-old congential *Nedd4-2^−/−^* mice. Scale bars, 50 µm and 10 µm (inset). (**e**) Summary of numeric densities of degenerative airway epithelial cells in congenital *Nedd4-2^−/−^* and control mice. *n* = 5–10 mice per group. * *p* < 0.05, ** *p* < 0.01, *** *p* < 0.001. Data are shown as mean ± S.E.M.

**Figure 6 ijms-22-06146-f006:**
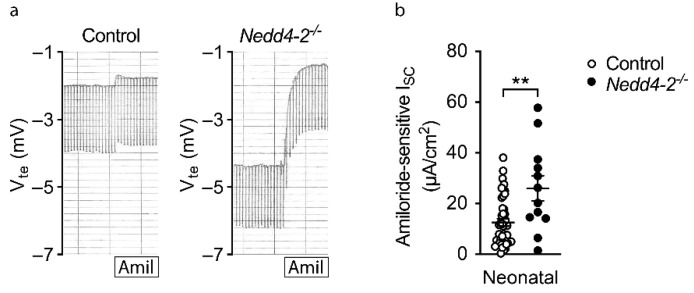
Increased ENaC activity in congenital *Nedd4-2^−/−^* mice: (**a**) Representative Ussing chamber recordings of the effect of amiloride (Amil) on transepithelial voltage (V_te_) and resistance (R_te_) of freshly excised tracheal tissues from a 10-day-old congenital *Nedd4-2^−/−^* mouse and a littermate control. (**b**) Summary of amiloride-sensitive short circuit current (I_SC_) across freshly excised airway tissues of 10-day-old congenital *Nedd4-2^−/−^* and control mice. *n* = 12–43 mice per group. ** *p* < 0.01. Data are shown as mean ± S.E.M.

**Figure 7 ijms-22-06146-f007:**
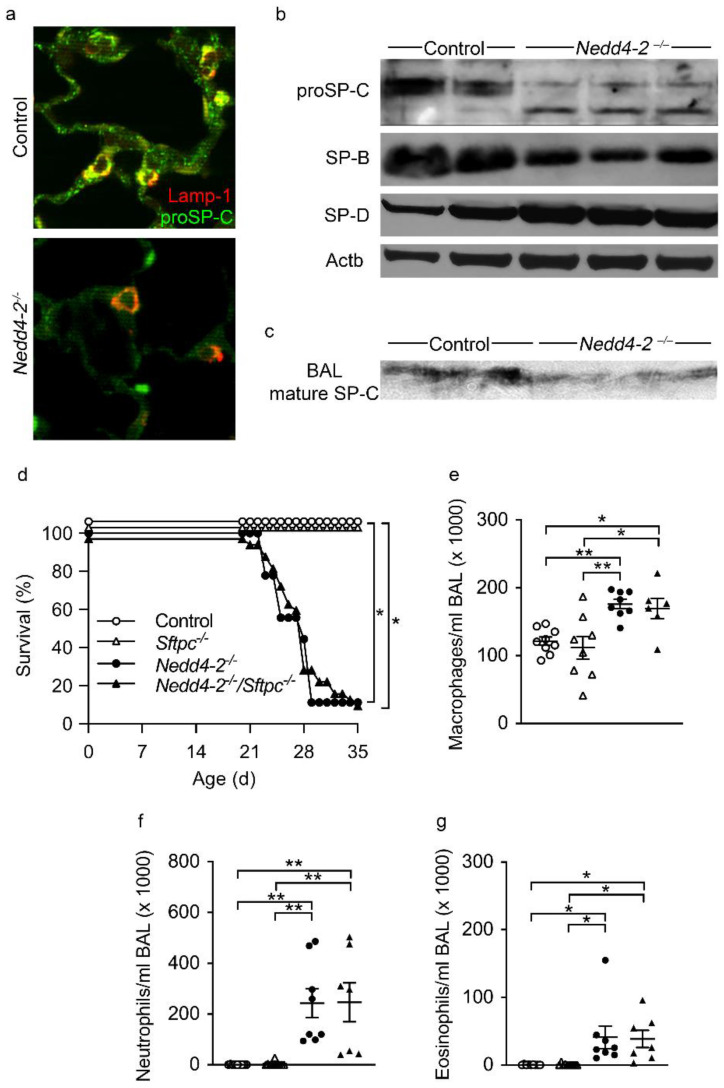
Evidence of proSP-C mistrafficking in lung epithelial cells of congenital *Nedd4-2^−/−^* mice: (**a**) Representative confocal images of double label fluorescence immunohistochemistry for proSP-C (green) and Lamp-1 (red). (**b**) Western blots for proSP-C, SP-B and SP-D from lung homogenates of 3-week-old congenital *Nedd4-2^−/−^* and control mice. (**c**) Western blots for mature SP-C in BAL of 3-week-old congenital *Nedd4-2^−/−^* and control mice. (**d**) Survival curve of congenital *Nedd4-2^−/−^*, congenital *Nedd4-2^−/−^/Sftpc^−/−^*, littermate *Sftpc^−/−^* and control mice. *n* = 9–32 mice per group. * *p* < 0.05. (**e**–**g**) Number of macrophages (**e**), neutrophils (**f**) and eosinophils (**g**) in BAL of 3-week-old congenital *Nedd4-2^−/−^*, congenital *Nedd4-2^−/−^/Sftpc^−/−^*, littermate *Sftpc^−/−^* and control mice. *n* = 7–9 mice per group. * *p* < 0.05, ** *p* < 0.01. Data are shown as mean ± S.E.M.

## Data Availability

The data presented in this study are available on request from the corresponding author.

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
