# Peer review of "Congenital Deletion of Nedd4-2 in Lung Epithelial Cells Causes Progressive Alveolitis and Pulmonary Fibrosis in Neonatal Mice"

_ijms, 2021, doi:10.3390/ijms22116146_

Round 1

Reviewer 1 Report

The manuscript by Leitz et al. describes the phenotype of mice with a congenital conditional loss of Nedd4-2 (NEDD4L) in club cells of the conducting airways and in type II pneumocytes mediated by the CCSP-rtTAS2-MS2/LCI construct.  Similar to their previous studies using the same cre deleter in adult animals, the authors report here that the congenital Nedd4-2 knockouts display a progressive lung disorder with hypoxemia, failure to thrive, sterile pneumonitis, fibrotic remodeling, and a high degree of lethality.  This was associated with increased expression of Muc5b and mucus plugging of distal airways (which was not evidenced in adult loss of function animals), increased ENaC activity, and mistrafficking of proSP-C. 

This manuscript was well written and the experiments were well designed and executed.  However, my main criticism of the study is that the authors fail to consider the important variable of sex in their analysis.  There are inherent sex differences in human lung structure and function that are apparent throughout the lifespan of individuals (emerging as early as in utero, thus influencing the process of lung development).  Further, male sex has been associated with decreased alveolar epithelium sodium transport capacity in rats (https://doi.org/10.1371/journal.pone.0136178).  Considering that the authors posit ENaC function to be critical to the emergence of the described phenotype, it seems necessary to consider this important parameter in the data analysis.  How many males and females were included in each group?  Were the groups sufficiently powered to detect any gender differences?  At the very least, this limitation must be acknowledged and discussed. 

On a more minor note, the authors might be interested to examine reported constraint metrics for the NEDD4L gene in the gnomAD database as it is reported to be highly loss of function intolerant.  While presumable gain of function NEDD4L variants have been previously associated with periventricular nodular heterotopia 7, homozygous loss of function variants have not been reported in the population.  The loss of function data from gnomAD combined with the data presented here as well as in previous genetically engineered mouse models might indicate that this is because that phenotype might promote early fetal or neonatal demise.  This too may be worth exploring further in the discussion as it might provide future testable hypothesis for chILD.

Reviewer 2 Report

    In this study, authors investigated a new mouse model of congenital deletion of Nedd4-2 in epithelial cells of the neonatal mouse lung that shares key aspects of interstitial lung diseases in children (chILD). This study is the continuous study of recent published work by same authors -“
Conditional deletion of Nedd4-2 in lung epithelial cells causes progressive pulmonary fibrosis in adult mice. 2020 Apr 24;11(1):2012. doi: 10.1038/s41467-020-15743-6.” and described in detail of Nedd4-2 knock down in lung epithelial cells in neonatal animal as a chilD disease model. The results are interesting and experiments were well performed. 

Minor comments:

  1. Is there any body size difference between congenital deletion of Nedd4-2 in epithelial cells neonatal mouse compare to control ?
  2. Figure 1a.b.c: group labels are missing in figures. Open dot= ? and Black dot=?
  3. Line 123: Figure A1 is mislabeled.
  4. Figure 2 legend: experimental animal numbers in each group are missing.
  5. Lane 140: numbers of neutrophils and eosinophils result is shown in figure 3e,f .
  6. Figure 7c: WB result is not clear even each lane of samples. I think need to replace with representative better image.
